# Association of Occupational Distress and Low Sleep Quality with Syncope, Presyncope, and Falls in Workers

**DOI:** 10.3390/ijerph182312283

**Published:** 2021-11-23

**Authors:** Nicola Magnavita, Reparata Rosa Di Prinzio, Gabriele Arnesano, Anna Cerrina, Maddalena Gabriele, Sergio Garbarino, Martina Gasbarri, Angela Iuliano, Marcella Labella, Carmela Matera, Igor Mauro, Franca Barbic

**Affiliations:** 1Postgraduate School of Occupational Health, Università Cattolica del Sacro Cuore, 00168 Rome, Italy; repdip@gmail.com (R.R.D.P.); gabrielearnesano93@gmail.com (G.A.); sgarbarino.neuro@gmail.com (S.G.); iuliano_angela@yahoo.it (A.I.); mau.igor91@yahoo.it (I.M.); franca.barbic@hunimed.eu (F.B.); 2Department of Woman, Child & Public Health Sciences, Fondazione Policlinico Universitario A. Gemelli IRCCS, 00168 Rome, Italy; 3Local Sanitary Unit Roma4, 00053 Civitavecchia, Italy; anna.cerrina@aslroma4.it (A.C.); maddalena.gabriele@aslroma4.it (M.G.); martina.gasbarri@aslroma4.it (M.G.); marcella.labella@aslroma4.it (M.L.); carmela.matera@aslroma4.it (C.M.); 4Department of Neuroscience, Rehabilitation, Ophthalmology, Genetics and Maternal-Infantile Sciences (DINOGMI), 16132 Genoa, Italy; 5Department of Biomedical Sciences, Humanitas University, 20072 Pieve Emanuele, Italy; 6Internal Medicine, IRCCS Humanitas Research Hospital, 20089 Rozzano, Italy

**Keywords:** loss of consciousness, mental health, working life, effort reward imbalance, sleep dis-orders, health promotion, workplace

## Abstract

Syncope and presyncope episodes that occur during work could affect one’s safety and impair occupational performance. Few data are available regarding the prevalence of these events among workers. The possible role of sleep quality, mental stress, and metabolic disorders in promoting syncope, presyncope, and falls in workers is unknown. In the present study, 741 workers (male 35.4%; mean age 47 ± 11 years), employed at different companies, underwent clinical evaluation and blood tests, and completed questionnaires to assess sleep quality, occupational distress, and mental disorders. The occurrence of syncope, presyncope, and unexplained falls during working life was assessed via an ad hoc interview. The prevalence of syncope, presyncope, and falls of unknown origin was 13.9%, 27.0%, and 10.3%, respectively. The occurrence of syncope was associated with an increased risk of occupational distress (adjusted odds ratio aOR: 1.62, confidence intervals at 95%: 1.05–2.52), low sleep quality (aOR: 1.79 CI 95%: 1.16–2.77), and poor mental health (aOR: 2.43 CI 95%: 1.52–3.87). Presyncope was strongly associated with occupational distress (aOR: 1.77 CI 95%: 1.25–2.49), low sleep quality (aOR: 2.95 CI 95%: 2.08–4.18), and poor mental health (aOR: 2.61 CI 95%: 1.78–3.84), while no significant relationship was found between syncope or presyncope and metabolic syndrome. These results suggest that occupational health promotion interventions aimed at improving sleep quality, reducing stressors, and increasing worker resilience might reduce syncope and presyncope events in the working population.

## 1. Introduction

Syncope, which is defined as a transient loss of consciousness due to cerebral hypoperfusion, is characterized by a rapid onset, short duration, and complete, spontaneous recovery [1,2]. It could be the result of a reduction of cardiac output caused by serious cardiovascular disorders (5–10%), such as arrhythmia or structural heart disease, but it is more commonly due to neuro-mediated mechanisms, such as vasovagal syncope or syncope associated with orthostatic hypotension [1,2,3]. Interestingly, prolonged bed rest in various clinical conditions (e.g., after surgery or trauma) may increase the risk of syncope [4]. Syncope is also quite frequently associated with constitutional hypotension, and is often underestimated in young women [5].

A transient loss of consciousness (T-LOC) constitutes a potential hazard if it occurs in the workplace. More than 90% of T-LOCs are due to syncope, epileptic seizures, or psychogenic non-epileptic seizures [2,6,7]. A prodromal period characterized by symptoms, such as light-headedness, nausea, sweating, weakness, and visual disturbances, may indicate that syncope is imminent, and is defined as “presyncope” [8]. Lastly, unexplained falls may hide episodes of syncope or presyncope [9]. 

Recently, there has been increased interest in the management of workers suffering from syncope [2,10,11,12] due to the potentially fatal outcome for the individual, or third parties, especially when an episode occurs in a worker undertaking hazardous tasks. In a Danish nationwide cohort, syncope was associated with a 1.4-fold greater risk of occupational accidents and a 2-fold higher risk of termination of employment, compared with the general population workforce [13]. The incidence of syncope in the military has been estimated at 7.2 cases per 1000 person-years [14]. A Polish study found that 4.7% and 14.8% of operating room staff had experienced at least one episode of syncope and presyncope, respectively [15]. The occurrence of these problems is rarely reported to health services. The Framingham offspring study revealed that 44% of the participants with an episode of loss of consciousness failed to undergo medical evaluation [16] and that a much higher percentage was observed in the younger population. In the Netherlands, admission to the Emergency Department (ED) accounts for only about 1% of expected syncope episodes in the general population, indicating that syncope and presyncope are underestimated [17]. In EDs, the management of patients suffering from T-LOC (syncope included) presents a challenge for the emergency physician [1,2,10], since he/she must first of all exclude the most serious causes of syncope before discharging a patient. However, patients diagnosed with benign vasovagal syncope (the commonest type of syncope) often receive no explanation of the event. Consequently, they are left with a fear of syncope recurrence and a possible reduction in the quality of life. 

Syncope and presyncope are common during work life [12,18,19,20] and may be triggered by occupational tasks requiring prolonged standing, exposure to a hot climate, or frequent changes of posture [11]. Furthermore, both central and peripheral initiating factors, such as mental, visceral, visual, and orthostatic distress, potentially present in work environments, or during specific occupational tasks, may facilitate vasovagal syncope [21], particularly in hypersusceptible individuals. During working life, environmental stressors, including psychosocial stress, may also play a role in promoting vasovagal syncope (the commonest cause of syncope), by facilitating the mechanisms suggested by Mosqueda-Garcia [22]. Figure 1 illustrates the mechanisms that activate vasovagal syncope and the possible role of the occupational environment. Syncope recurs is about 22% of subjects at 2 years from the first episode [23]. Interestingly, the highest risk of syncope recurrence in the first 6 months after the first episode was observed in individuals of working age [12].

In Italy, in accordance with European directives, the employer must assess and monitor occupational risk factors for health and safety, provide training for workers on the occupational risks identified, and guarantee that the employees’ health is monitored by an occupational physician. The occupational physician, whose aim is to improve workers’ health, is also responsible for promptly identifying the symptoms of diseases that may interfere with work safety. The present study, which was performed during periodic medical examination in the workplace, aimed at quantifying the prevalence of syncope and presyncope occurring during the working life of individuals employed in different companies. We also assessed whether these episodes were associated with low sleep quality, occupational distress, and mental health. In addition, an evaluation was made in the same population of a possible association between metabolic syndrome and syncope and presyncope.

## 2. Materials and Methods

### 2.1. Study Population

All 754 workers consecutively enrolled for their periodic medical examination from July to December 2020 were asked to fill in an anamnestic questionnaire regarding the occurrence of syncope, presyncope, and falls (including those without any apparent cause) during their working life. The workers came from 10 companies and belonged to the health sector (57%), social and educational services (18%), industry (2%), and commerce (23%). Eight workers refused to participate in the survey; another five, who provided incomplete answers, were excluded from the study. 

All subjects gave their informed consent to the processing of personal data for research purposes. The research, which was conducted in compliance with the principles of the Helsinki Declaration, was part of the mandatory medical surveillance of workers in the workplace, and was approved by the University Ethics Committee.

### 2.2. Questionnaire

#### 2.2.1. Prevalence of Syncope, Presyncope, and Falls

To ascertain the prevalence of syncope, presyncope, and unexplained falls, the workers were asked to answer some questions. For syncope: “In your working life have you ever experienced a temporary loss of consciousness, fallen down, and then spontaneously recovered consciousness?”; for presyncope: “In your working life have you ever been on the verge of fainting or felt as if you were fainting with cold sweats, intense weakness, and the need to sit or lie down?”; for falls of unknown cause: “In your working life have you ever fallen to the ground?”; and: “Did the fall occur without a clear accidental reason (tripping, pushing, sliding, etc.)?”. The number and date of episodes, the presence of associated trauma, familiarity with similar episodes, and the occurrence of episodes during working activity were also assessed. These questions were added to a standard questionnaire used during periodic medical examination.

#### 2.2.2. Occupational Stress

Occupational stress was assessed using the Italian version [24,25] of the Siegrist effort/reward imbalance model [26,27]. The questionnaire contains 10 questions whose answers are graded according to a four-point Likert scale. The Effort sub-scale, which determines the psychological effort made to work, consists of three questions and the score ranges from 3 to 12. The reward sub-scale is composed of seven questions and the score ranges from 7 to 28. The weighted relationship between the two questions, effort/reward imbalance index (ERI) is conventionally considered an expression of distress if higher than one. We verified the reliability of the questionnaire by calculating Cronbach’s alpha of our data. This measure of internal consistency in our survey was equal to 0.822 (effort), and 0.710 (reward). Since Cronbach’s ranges from 0 to 1, the reliability of the questionnaire in this study was good/acceptable [28].

#### 2.2.3. Sleep Quality

The quality of sleep was assessed using the Italian version [29] of the Pittsburgh questionnaire [30], which consists of 18 questions that form the Pittsburgh Sleep Quality Index (PSQI). An overall score equal to, or greater than 5, corresponds to poor sleep quality (bad sleepers). Cronbach’s alpha in this study was 0.835 (good/excellent reliability).

#### 2.2.4. Mental Health

The mental health of workers was assessed using the Italian version [31,32] of the 12-item General Health Questionnaire (GHQ-12) [33,34], which is a reliable screening instrument for psychological distress and a measure of the common mental health problems/domains of depression, anxiety, somatic symptoms, and social withdrawal, rated on a four-point scale. We used the scoring method (0–0–1–1) with the cut-off level recommended by the authors [35]. A value ≥ 3 was classified as low mental health. The reliability of the questionnaire in this study was 0.868 (excellent).

### 2.3. Medical Examination and Blood Tests

The occupational physician in charge carried out a clinical evaluation of all the workers participating in the study. In addition, workers’ height and weight were recorded to quantify body mass index (BMI), and their waist circumference was measured at the narrowest point between the lower costal (10th rib) and the iliac crest. A sphygmomanometer was used to measure blood pressure while workers were supine, and during active standing. Venous fasting blood samples were collected in plain tubes and centrifuged at 3000 rpm for 10 min at room temperature, and serum samples were frozen at −20°C until assayed. Fasting blood glucose (FBG) and blood lipid profile (total cholesterol, high-density lipoprotein, triglycerides, and low-density lipoprotein) were determined using an enzymatic assay kit.

#### Metabolic Syndrome Prevalence

Components of metabolic syndrome were defined according to the International Diabetes Federation (IDF) [36], the National Cholesterol Education Program Expert Panel on Detection Evaluation and Treatment of High Cholesterol in Adults (NCEP/ATPIII) [37], and the American Association of Clinical Endocrinologists (AACE) [38]. Obesity was defined as BMI ≥25 kg/m^2^, or a waist circumference of ≥ 94 cm. for men and ≥ 80 cm for women, while hypertriglyceridemia was defined as a serum triglyceride level >150 mg/dL (1.7 mmol/L). A low level of high-density lipoprotein (HDL) serum cholesterol was defined as a serum HDL-cholesterol <40 mg/dL (1.03 mmol/L). A systolic blood pressure >130 mmHg, and/or a diastolic blood pressure >85 mmHg or drug treatment for hypertension were classified as high blood pressure, while a plasma glucose level >100 mg/dL (5.6 mmol/L) or the presence of hypoglycemic drug treatment were classified as high fasting glucose. The presence of three or more abnormalities in the aforementioned components was considered to constitute metabolic syndrome (MetS) [39].

### 2.4. Statistics

Sociodemographic features were analyzed using frequency or statistical distribution for categorical and continuous variable, respectively. In accordance with the literature [40], the age of 55 years was used as a cut-off to divide participants into older and younger workers. The Chi-square test was used to compare case distribution by sex and age, while the unpaired Student’s t test (for parametric data) and the Mann–Whitney U test (for ordinal data) were used to compare occupational stress, sleep quality, mental health, and metabolic syndrome prevalence (treated as continuous variables) in workers with and without syncope, presyncope, and unexplained falls. 

Using socio-demographics as correction factors, a logistic regression analysis was performed to ascertain if the occupational variables investigated could predict the occurrence of syncope, presyncope, and falls. The estimated effect was presented in terms of adjusted odds ratio (aOR) and 95% confidence intervals. Statistical analyses were performed using the IBM Statistical Package for Social Sciences, SPSS, version 26.0 statistical software (IBM, Armonk, NY, USA). The significance criterion for a two-tailed *p* value ≤ 0.05 was applied.

## 3. Results

Overall, 741 subjects classified as fit to work (male 262, 35.4%; female 479, 64.6%; mean age 47 ± 11 years) participated in the study, corresponding to 98.3% of the workers selected. Table 1 shows the prevalence of T-LOC episodes as well as stress, sleep problems, and metabolic syndrome in all workers, according to sex and age. More than 50% of workers reported at least one episode of syncope, presyncope, or falls of unknown origin during their period of employment. Syncope and presyncope were more frequent in women than in men. Syncope was reported more frequently in young workers than in those over 55 years of age. Falls were quite frequent, affecting more than one in three workers. Interestingly, unexplained falls resulting from no apparent cause affected 10% of the workers, with no sex or age differences. Occupational stress, sleep quality, and mental health showed no significant sex differences. Older workers reported bad quality of sleep, occupational stress, and psychological disorders more frequently than others. Metabolic syndrome was more frequent in males than in females, and in older rather than younger workers. 

Most workers reported only one syncope; 37 workers (35.9%) reported two to four episodes, and 13 (12.6%) reported five or more syncope recurrences. In 18 cases, syncope had occurred in the previous two years. Only 12 workers reported syncope occurrence during working activity. As indicated in Table 2, workers who reported syncope or presyncope were found to have a lower quality of sleep, a reduced level of mental health, and greater occupational distress than the other workers. The mean scores of workers with recurrent syncope differed the most from the mean values of the group. Workers reporting falls of unknown origin had lower levels of sleep quality than the group, but did not report higher occupational stress (Table 2).

The levels of HDL-cholesterol triglycerides, blood glucose, and blood pressure in workers with one or more T-LOC episodes did not differ from those measured in other workers (Table 3). Workers with one or more syncopal episodes had a significantly lower than average BMI. On the other hand, workers with falls of unknown origin had a higher than average BMI (Table 3).

Table 4 indicates the results of logistic regression in models adjusted for age and sex. Interestingly, syncope and presyncope were associated with an increased risk of occupational distress, sleep problems and poor mental health, whereas no association was found between syncope, presyncope, or falls without apparent cause, and a diagnosis of metabolic syndrome (Table 4).

## 4. Discussion

This study, which, to the best of our knowledge, is the first conducted in the workplace to assess the prevalence of syncope, presyncope, or falls with no apparent cause, indicates that these phenomena are quite common, and could occur during the performance of work activities, when they could potentially endanger both the safety of the worker and that of colleagues or clients, and also affect the continuity of the production process.

In our study, loss of consciousness involved about one worker in seven, with a greater frequency in females and younger workers. Presyncope symptoms were much more common, affecting more than one in four workers, and falls for no apparent reason affected one in ten. In our sample, lifetime prevalence was lower than that observed in the general population, where it is estimated to be 42%, with a higher percentage between 10 and 30 years of age, mainly of vasovagal syncope [1,8,41]. In occupational cohorts, prevalence rates have been reported to be as high as 35% in healthcare workers [20], 39% among medical students [42], and 41% in air force employees (with recurrent syncope occurring in 13.5%) [43]. The disparity between our data and those reported in the general population may be explained by the structure of the age groups in our sample. Syncope typically follows a tri-modal distribution in both sexes, with an increase in cases before 20, around 60, and over 80 years of age [44,45]. In our cohort, the third modal peak was missing and the first was based on few participants. Most of the episodes had occurred in younger workers, while those reported by middle-aged workers had often occurred many years previously. The higher prevalence of syncopal episodes in women corresponded to data reported in the literature [46]. Women are younger than men at the time of their first syncope, have lower baseline systolic blood pressure (according to data reported in patients with constitutional hypotension) [5], experience heat as a more common trigger, have physical symptomatology more frequently (e.g., feeling warm, seizures, and greater post-syncope fatigue), and are more prone to recurrent syncope [47]. These differences, in terms of age and clinical presentation, may explain the higher prevalence in females in this occupational cohort.

In our cohort, syncope and presyncope were significantly associated with occupational distress, low quality of sleep, and poor mental health, while no association with metabolic risk factors and cardiovascular risk was observed. Based on these epidemiological findings, it appears that the syncopal and presyncope episodes observed in active workers were predominantly attributable to neuro-mediated mechanisms rather than severe cardiac conditions. Our study data are in keeping with the hypothesis of central mechanisms involved in the reflex syncope reported by Mosqueda-Garcia [22], which focuses specifically on the role of the occupational environment (Figure 1). In fact, workers are a special category in which the presence of serious heart disease can lead to removal from occupational risk and early retirement. This phenomenon, known as the “healthy worker effect”, causes a prevalence of severe heart disease that is lower in the workforce than in the general population. Nonetheless, we cannot rule out that some of the reported episodes may have been caused by heart disease. For this reason, all workers who reported syncopal episodes were advised to see their General Practitioner to obtain a definite diagnosis, in accordance with National Health plan indications. 

The sudden and brief loss of consciousness that many workers reported may have been due to a number of different diseases, such as orthostatic hypotension (caused by drugs, hypovolemia, primary or secondary autonomic failure, others), neurally-mediated syncope, cardiogenic syncope, or less frequently, to other neurologic disorders such as epilepsy, psychogenic syncope, vertebrobasilar transient ischemic attacks), metabolic disorders, and intoxication [8,48]. According to the literature, vasovagal is the commonest form of syncope [49]. It accounts for 60–80% of cases of syncope, and typically occurs in young adults [4,50]. The absence of structural heart disease in our sample and the higher prevalence in younger rather than older workers suggest that most of the reported episodes may be attributed to this benign form. Although vasovagal syndrome is generally a benign condition from a clinical point of view, it may have a negative impact on work. Indeed, it may create a danger for workers and third parties [11] if it occurs during highly hazardous tasks, and may result in a significantly increased economic burden. Recurrent cases of syncope may reduce the quality of life and promote occupational injury [11]. Furthermore, they are associated with an increased risk of death and major adverse cardiovascular events [51,52]. Atypical vasovagal syncope (commoner in older adults) and non-neurogenic syncope can often be erroneously misdiagnosed as falls. We agree with Kenny [49] that a more standardized approach should be adopted in the diagnosis and management of workers presenting with syncope or unexplained falls.

In this study, we observed an association between syncope occurrence and occupational stress. A previous study reported that emotional vasovagal syncope might be associated with distress [53]. In addition, a longitudinal study showed that psychosocial impairment reliably predicted non-response to treatment of syncope [54]. This finding supports the hypothesis that psychosocial occupational stress factors may have an important role in syncope occurrence, particularly in hypersusceptible individuals. 

The association we observed between syncope and low sleep quality was not completely unexpected. Indeed, daytime sleepiness due to low quality of night sleep has been associated with orthostatic hypotension that may promote syncope, presyncope, and falls [55]. Sporadic observations in the literature suggest a link between sleep apnea syndrome and vasovagal syncope [56]. Interestingly, sleep quality is a mediator between occupational stress and its effects on metabolic pathologies [57] and mental health [58]. Vasovagal syncope has also been previously associated with mental health problems [59]. In a longitudinal study, syncope patients exhibited high levels of psychological distress and mood/anxiety disorders [60]. Psychiatric disorders are common in patients with tilt-induced vasovagal syncope, and seem to predict the risk of recurrence [59].

A significant share of the workers in our sample had recurrent episodes of syncope. Considerable evidence shows that as in other chronic diseases, syncope recurrence affects the quality of life [61] by impacting negatively on numerous daily activities, such as driving, working, and attending school. It also impairs mental health with episodes of somatization, depression, and anxiety [62,63,64].

Investigating syncope while visiting the workplace is a useful and not excessively time-consuming activity. During his/her visit, the occupational physician can investigate the history of syncope episodes by asking whether the episode happened at work, whether there was any injury, whether there were other cases in the family, and what action the worker took to clarify the source of the problem. The doctor can also measure the worker’s blood pressure in both a supine and standing position to evaluate orthostatic syncope. These activities take a few minutes and significantly improve the quality of care provided in the workplace. If the episode is recent, the occupational health physician should consider whether syncope would deeply affect the worker’s daily life. In cases where the work setting is intrinsically hazardous, the physician must evaluate whether the worker can safely go on working and driving. A quantitative model has been proposed [11] for guiding the physician in stratifying the risk for workers who have had a previous syncope event. This model considers the risk of syncope recurrence, job task duration, and characteristics that facilitate a syncope during work.

Our study has some limitations. The use of a convenience sample suggests a very cautious application of the results in other occupational situations. However, our research has shown that investigating T-LOC events among workers is a useful activity that does not put an excessive burden on health and safety services, and could be carried out with a limited use of resources. Another limitation is the cross-sectional nature of our study, which does not allow us to infer causality. Furthermore, since the episodes of T-LOC are self-reported, we cannot exclude under-reporting or over-reporting.

## 5. Conclusions

The association of symptoms with excessive occupational stress, low quality of sleep, and mental balance disturbances indicates the need to consider health promotion intervention in the workplace. A prevention program should aim at improving sleep quality, reducing stressors, and increasing worker resilience. Psychological support should be provided, at least for workers with more evident mental health problems.

## Figures and Tables

**Figure 1 ijerph-18-12283-f001:**
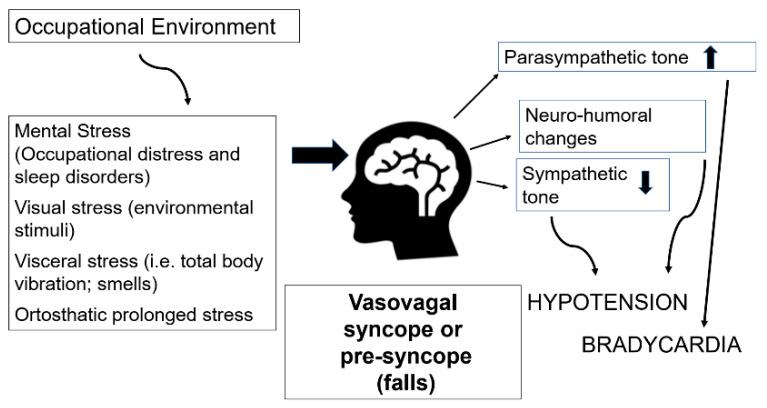
Potential role of environmental stressors in promoting vasovagal syncope (from Mosqueda-Garcia [22], modified).

**Table 1 ijerph-18-12283-t001:** Prevalence of syncope, presyncope and falls, and distribution of occupational distress, low quality of sleep, impaired mental health, and metabolic syndrome according to sex and age.

Type of Problem	Total Number (%) 741 (100)	Male Number (%) 262 (35.4)	Female Number (%) 479 (64.6)	Chi-Square *p*	Younger ^4^ Number (%) 526 (71.0)	Older ^5^ Number (%) 215 (29.0)	Chi-Square *p*
Syncope	103 (13.9)	70 (7.6)	83 (17.3)	0.000	84 (16.0)	19 (8.8)	0.011
Presyncope	200 (27.0)	48 (18.3)	152 (31.7/	0.000	151 (28.7)	49 (22.8)	0.100
Fall	272 (36.7)	92(35.1)	180 (37.6)	0.506	185 (35.2)	87 (40.5)	0.175
Fall unknown origin	76 (10.3)	22 (8.4)	54 (11.3)	0.217	53 (10.1)	23 (10.7)	0.800
Distressed ^1^	278 (38.6)	110 (42.5)	168 (36.4)	0.106	182 (35.6)	96 (45.7)	0.011
Bad sleeper ^2^	360 (48.6)	120 (45.8)	240 (50.1)	0.263	234 (44.5)	126 (58.6)	0.001
Low mental health ^3^	152 (20.5)	51 (19.5)	101 (21.1)	0.592	97 (18.5)	55 (25.6)	0.030
Metabolic syndrome	91 (12.3)	46 (17.6)	45 (9.4)	0.001	43 (8.2)	48 (22.3)	0.001

**Notes.** ^1^ Effort reward imbalance ERI > 1; ^2^ Pittsburgh Sleep Quality Index PSQI ≥ 5; ^3^ General Health Questionnaire GHQ12 ≥ 3; ^4^ age < 55 years; ^5^ age ≥ 55 years.

**Table 2 ijerph-18-12283-t002:** Comparison of mean values of stress, sleep quality, and mental health in workers with or without syncope, presyncope and falls of unknown origin. (Mann–Whitney U test).

Type of Problem	Stress (ERI)	Sleep Quality (PSQI)	Mental Health (GHQ-12)
Syncope	1.05 ± 0.46 vs. 0.90 ± 0.43 ***	6.62 ± 4.32 vs. 4.81 ± 3.23 ***	2.48 ± 3.32 vs. 1.33 ± 2.32 ***
Recurrent syncope	1.15 ± 0.51 vs. 0.91 ± 0.42 ***	7.72 ± 4.54 vs. 4.87 ± 3.28 ***	3.34 ± 3.89 vs. 1.36 ± 2.33 ***
Recent syncope	1.17 ± 0.47 vs. 0.92 ± 0.43 *	7.56 ± 5.01 vs. 5.00 ± 3.39 *	2.67 ± 2.97 vs. 1.46 ± 2.49 ***
Presyncope	1.06 ± 0.46 vs. 0.88 ± 0.41 ***	6.88 ± 3.97 vs. 4.39 ± 2.98 ***	2.40 ± 3.21 vs. 1.16 ± 2.10 ***
Fall of unknown cause	0.97 ± 0.46 vs. 0.92 ± 0.43	6.22 ± 4.28 vs. 4.93 ± 3.33 **	2.33 ± 3.57 vs. 1.40 ± 2.34

**Notes.** * *p* < 0.05; ** *p* < 0.01; *** *p* < 0.001.

**Table 3 ijerph-18-12283-t003:** Comparison of mean values of systolic and diastolic blood pressure, HDL-cholesterol, triglycerides, blood glucose, and BMI in workers with or without T-LOC. (Student’s t test).

Type of Problem	Systolic Blood Pressure	Diastolic Blood Pressure	HDL Cholesterol
Syncope	122.35 ± 16.40 vs. 122.28 ± 16.29	79.09 ± 13.97 vs. 79.00 ± 11.96	63.86 ± 15.48 vs. 61.96 ± 16.22
Recurrent syncope	122.00 ± 16.74 vs. 122.31 ± 16.27	77.77 ± 12.26 vs. 79.11 ± 12.24	65.96 ± 15.64 vs. 61.89 ± 16.13
Presyncope	121.66 ± 16.63 vs. 122.51 ± 16.19	79.34 ± 13.20 vs. 79.90 ± 11.91	65.96 ± 15.64 vs. 61.89 ± 16.13
Fall of unknown cause	119.59 ± 18.17 vs. 122.58 ± 16.07	77.84 ± 13.92 vs. 79.13 ± 12.06	59.06 ± 16.56 vs. 62.63 ± 16.04
**Type of Problem**	**Triglycerides**	**Blood Glucose**	**BMI**
Syncope	93.02 ± 54.72 vs. 102.67 ± 54.26	87.41 ± 16.43 vs. 91.14 ± 14.23	23.25 ± 3.32 vs. 25.03 ± 4.38 ***
Recurrent syncope	86.85 ± 60.70 vs. 102.59 ± 53.65	86.84 ± 21.42 vs. 90.92 ± 13.86	24.10 ± 4.03 vs. 24.82 ± 4.30
Presyncope	98.26 ± 52.51 vs. 102.41 ± 55.09	88.53 ± 14.53 vs. 91.36 ± 14.59	24.55 ± 4.57 vs. 24.85 ± 4.17
Fall of unknown cause	114.85 ± 50.77 vs. 99.74 ± 54.64	88.97 ± 15.39 vs. 90.79 ± 14.52	25.87 ± 5.04 vs. 24.65 ± 4.18 *

**Notes.** * *p* < 0.05; *** *p* < 0.001.

**Table 4 ijerph-18-12283-t004:** Association of episodes of syncope, presyncope and falls with occupational distress, poor sleep quality, impaired mental health, and metabolic syndrome (logistic regression models adjusted by age and sex).

Type of Problem	Distress OR (CI95%)	Bad Sleep OR (CI95%)	Low Mental Health OR (CI95%)	MetS OR (CI95%)
Syncope	1.62 (1.05; 2.52) *	1.79 (1.16; 2.77) ***	2.43 (1.52; 3.87) ***	0.61 (0.27; 1.39)
Recurrent syncope	2.11 (1.15; 3.88) *	2.19 (1.18; 4.04) *	3.88 (2.12; 7.08) ***	1.02 (0.40; 2.75)
Recent syncope	1.89 (0.71; 5.04)	1.68 (0.63; 4.45)	2.55 (0.96; 6.78)	1.81 (0.48; 6.78)
Presyncope	1.77 (1.25; 2.49) ***	2.95 (2.08; 4.18) ***	2.61 (1.78; 3.84) ***	1.21 (0.71; 2.04)
Fall unknown cause	1.00 (0.61; 1.66)	1.49 (0.91; 2.42)	1.69 (0.99; 2.87)	1.33 (0.67; 2.65)

**Notes.** OR: odds ratio; CI95%: confidence interval at 95%; distress: ERI > 1; bad sleep: PSQI ≥ 5; low mental health: GHQ-12 ≥ 3; MetS: metabolic syndrome, three or more components (hypertension, hyperglycemia, low HDH cholesterol, hypertriglyceridemia, obesity). * *p* < 0.05; *** *p* < 0.001.

## Data Availability

The data presented in this study are freely available on Zenodo repository.

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
