# Peer review of "Association of Occupational Distress and Low Sleep Quality with Syncope, Presyncope, and Falls in Workers"

_ijerph, 2021, doi:10.3390/ijerph182312283_

Round 1

Reviewer 1 Report

The authors do a great job of concisely summarizing the relationship between workers' health symptoms and risk for on-the-job falls. This study was well-designed and analyzed. More importantly, it addresses an important issue-- worker safety and well-being. The authors are to be commended; this paper is refreshing and insightful.  The story is clear, but there are some technical errors throughout (see comments below).

The last author's name is missing.

The term "syncope index" is not clearly defined. Please rephrase the first paragraph on Page 3 be more clear. 

The paragraph formatting needs to be fixed throughout. Paragraphs should contain more than one sentence, but should not be a full page long.

Fix Table 1 "gen-der" typo.

Please check the manuscript throughout for grammatical syntax errors. The placement of commas is not consistent with English writing. This makes it difficult to read the manuscript or interpret results at points.

Author Response

Reviewer #1

The authors do a great job of concisely summarizing the relationship between workers' health symptoms and risk for on-the-job falls. This study was well-designed and analyzed. More importantly, it addresses an important issue-- worker safety and well-being. The authors are to be commended; this paper is refreshing and insightful.  The story is clear, but there are some technical errors throughout (see comments below).

Response:

We thank the reviewer for the kind attention with which he/she considered our manuscript and for the appreciation that encourages us to continue our work.

The last author's name is missing.

Response:

We thank the reviewer for having found this typesetting error. Indeed, the last author is Franca Barbic. We have now corrected the typo.

The term "syncope index" is not clearly defined. Please rephrase the first paragraph on Page 3 be more clear.

Response:

We agree with the reviewer. The term "syncope index" may be misleading and be confused with the syncope-falls index [Fitzpatrick N, Romero-Ortuno R. The Syncope-Falls Index (SYFI): A tool for predicting risk of syncope and complex falls in the older adult based on cumulative health deficits. QJM. 2021 May 20:hcab141. doi: 10.1093/qjmed/hcab141.] We believe the term "the first syncope episode" is clearer. In the revised manuscript we have changed the term “syncope index” to “first syncope episode”.

The paragraph formatting needs to be fixed throughout. Paragraphs should contain more than one sentence, but should not be a full page long.

Response:

We have gladly accepted this suggestion. In the revised manuscript we have arranged the paragraphs as requested.

Fix Table 1 "gen-der" typo.

Response:

We have fixed the error in the revised Table1.

Please check the manuscript throughout for grammatical syntax errors. The placement of commas is not consistent with English writing. This makes it difficult to read the manuscript or interpret results at points.

Response:

We apologize for this. Indeed, some errors arose in the typesetting of the paper. We have arranged for an additional correction of the revised version of the manuscript by a native English speaker.

Reviewer 2 Report

The study is well designed. It is exciting and straightforward to read within its limitations of dealing with a specific working population. Several small changes are necessary to improve the quality of the work.

The material and methods section is unclear which sector the workers belong to agriculture, industry, transport, health personnel, etc. This aspect should be specified.

Reading the article, the high prevalence of syncope is striking, so it is essential, as mentioned above, to indicate the population in which the study is being conducted.

It would be interesting to provide a small paragraph with information on the occupational health system in Italy to understand in what context the study has been conducted.

The terms gender and sex should be used appropriately. There is a tendency to use the word gender instead of sex because it seems more politically correct. However, this is a mistake from a scientific point of view and can lead to misinterpretation of the data. When we have information about the sexual orientation of a person, e.g., gay, non-binary, lesbian, bisexual, transgender, we should talk about gender. If they do not have the information on sexual orientation and refer to biological characteristics, then one should speak of sex. As the authors use the word gender, they should specify that it is collected the sexual orientation of the participants or otherwise change the word sex.

The material and methods indicated that the authors used the Italian version of the Pittsburgh Sleep Quality Index (PSQI), and Cronbach's alpha statistic is provided. It is unclear whether the alpha statistic refers to the validity of the Italian version of the scale or whether the authors have calculated it from their data. This issue of the Cronbach alpha should be explained in detail. If the authors have calculated the alpha from the data collected in their questionnaires, the calculations should be explained more in the results section. What are the implications of this alpha value being high or low?

The same is true for the Goldberg General Health Questionnaire (GHQ): it is indicated that the Italian version of the GHQ was used, and at the end of the paragraph, the reliability value is displayed. We do not know whether the reliability refers to the Italian version of the questionnaire or whether the article's authors measured it. On the other hand, the term reliability can be misleading because it could be the internal consistency measured by Cronbach's Alpha, the system of two halves, or the Parallel-forms method, Test-retest reliability method:

In table 1 in the column of the probability of the chi-square shows the value 0.0000. It is incorrect, and it should be replaced by > 0.0001.

In table 2, a Student's t-test has been calculated. However, the data are scales. It is more appropriate to perform a nonparametric test,  the U-Mann Witney test, on this table.

In the article, the authors indicate that the problem of syncope could be addressed, and preventive measures could be taken with few resources. It would be interesting if, in the discussion, they could comment on how much additional time would be necessary for each health examination to study syncope and the associated factors.

Author Response

Reviewer #2

The study is well designed. It is exciting and straightforward to read within its limitations of dealing with a specific working population. Several small changes are necessary to improve the quality of the work.

Response: We thank the reviewer for having appreciated the present study and for providing useful suggestions to improve the manuscript.

The material and methods section is unclear which sector the workers belong to agriculture, industry, transport, health personnel, etc. This aspect should be specified. Reading the article, the high prevalence of syncope is striking, so it is essential, as mentioned above, to indicate the population in which the study is being conducted.

Response:

The workers enrolled in the present study came from 10 different companies and belonged to the health, social and educational services, industry, and tertiary sector. We have added this information in the method section (lines 112-114), as it follows: “The workers came from 10 companies and belonged to the health sector (57%), social and educational services (18%), industry (2%), and commerce (23%).”

It would be interesting to provide a small paragraph with information on the occupational health system in Italy to understand in what context the study has been conducted.

Response:

We thank the reviewer for this suggestion that enables us to furnish more details on the organization of the occupational health system in Italy. In Italy, in accordance with European directives, the employer must assess and monitor occupational risk factors for health and safety, provide training for workers on the occupational risks identified, and guarantee that the employees’ health is monitored by an occupational physician. In the revised version of the manuscript, we have added a new paragraph that gives this information (Lines 95-98).

The terms gender and sex should be used appropriately. There is a tendency to use the word gender instead of sex because it seems more politically correct. However, this is a mistake from a scientific point of view and can lead to misinterpretation of the data. When we have information about the sexual orientation of a person, e.g., gay, non-binary, lesbian, bisexual, transgender, we should talk about gender. If they do not have the information on sexual orientation and refer to biological characteristics, then one should speak of sex. As the authors use the word gender, they should specify that it is collected the sexual orientation of the participants or otherwise change the word sex.

Response:

We completely agree with this approach. We have changed “gender” into “sex” throughout the manuscript. Thank you for this important observation.

The material and methods indicated that the authors used the Italian version of the Pittsburgh Sleep Quality Index (PSQI), and Cronbach's alpha statistic is provided. It is unclear whether the alpha statistic refers to the validity of the Italian version of the scale or whether the authors have calculated it from their data. This issue of the Cronbach alpha should be explained in detail. If the authors have calculated the alpha from the data collected in their questionnaires, the calculations should be explained more in the results section. What are the implications of this alpha value being high or low?

Response:

Cronbach’s alpha is a measure of internal consistency, and it is used as a measure of scale reliability. All the tests used in this research were validated in Italian and therefore, during the validation studies that are in our references, the authors calculated the respective alpha of the tests. We wanted to verify the results obtained in our series, not only to confirm the reliability of the test (which was already recognized), but also to have an indirect measure of the correctness of the answers. In social sciences, as a rule of thumb, the internal consistency of a test is considered poor if alpha ranges between 0.5 and 0.6, questionable (>0.6), acceptable (>0.7), good (>0.8), excellent (>0.9) [Tavakol M, Dennick R. Making sense of Cronbach's alpha. Int J Med Educ. 2011;2:53-55. doi: 10.5116/ijme.4dfb.8dfd.] We have added a few notes on this issue in the text (Page… Lines…)

The same is true for the Goldberg General Health Questionnaire (GHQ): it is indicated that the Italian version of the GHQ was used, and at the end of the paragraph, the reliability value is displayed. We do not know whether the reliability refers to the Italian version of the questionnaire or whether the article's authors measured it. On the other hand, the term reliability can be misleading because it could be the internal consistency measured by Cronbach's Alpha, the system of two halves, or the Parallel-forms method, Test-retest reliability method:

Response:

Also in this case, we referred the measure of internal consistency drawn from our data. We have specified that the value was measured in our data. We agree that the term reliability can be misleading.

In table 1 in the column of the probability of the chi-square shows the value 0.0000. It is incorrect, and it should be replaced by > 0.0001.

Response:

True! We have changed 0.0000 to < 0.0001 in Table 1. Thank you for this comment.

In table 2, a Student's t-test has been calculated. However, the data are scales. It is more appropriate to perform a nonparametric test,  the U-Mann Witney test, on this table.

Response:

We thank the reviewer for this constructive criticism on the statistical methods. We have re-analysed the data by using the non-parametric Mann-Whitney U test. Table 2 has been revised according to the new statistical analyses. We have modified the Material and Methods section ( 2.4 Statistics page… line…) as requested.

In the article, the authors indicate that the problem of syncope could be addressed, and preventive measures could be taken with few resources. It would be interesting if, in the discussion, they could comment on how much additional time would be necessary for each health examination to study syncope and the associated factors.

Response:

We thank the reviewer for having underlined his/her interest in this crucial preventive issue that can be addressed in the workplace. Indeed, as already reported by Barbic et al (Auton. Neuroscience) 2014), during the workers’ medical examination, it is important to investigate a possible history of syncope, pre-syncope, or falls of unknown origin, especially among workers engaged in extremely risky jobs. In addition, the measurement of blood pressure while workers are supine and also during standing, may reveal the presence of an orthostatic hypotension, possibly of iatrogenic origin, that may result in orthostatic syncope or presyncope during work. We believe that this issue may be addressed during health surveillance in the workplace and may require no more than 5-10 minutes if the physician and/or the occupational nurse are trained to do this. We have included this additional information in the discussion as requested. In our experience, these activities take a few minutes and significantly improve the quality of care provided in the workplace. The lines 333-346 now are as it follows: “Investigating syncope while visiting the workplace is a useful and not excessively time-consuming activity. During his/her visit, the occupational physician can investigate the history of syncope episodes by asking whether the episode happened at work, whether there was any injury, whether there were other cases in the family, and what action the worker took to clarify the source of the problem. The doctor can also measure the worker’s blood pressure in both a supine and standing position to evaluate orthostatic syncope. These activities take a few minutes and significantly improve the quality of care provided in the workplace. If the episode is recent, the occupational health physician should consider whether syncope will deeply affect the worker’s daily life. In cases where the work setting is intrinsically hazardous, the physician must evaluate whether the worker can safely go on working and driving. A quantitative model has been proposed [11] for guiding the physician in stratifying the risk for workers who have had a previous syncope event. This model considers the risk of syncope recurrence, job task duration, and characteristics that facilitate a syncope during work.”

Reviewer 3 Report

thank you for reading the article. Very interesting article. I propose to complete the research limitations as well as therapeutic implications in the article. I suggest that you also use the publication by Krupa S. Sleep disorders among nurses and other health care workers in Poland during the COVID-19 pandemic, which also describes problems during the epidemic. 

Author Response

Reviewer #3

thank you for reading the article. Very interesting article. I propose to complete the research limitations as well as therapeutic implications in the article. I suggest that you also use the publication by Krupa S. Sleep disorders among nurses and other health care workers in Poland during the COVID-19 pandemic, which also describes problems during the epidemic.

R.: We have fully appreciated this article that describes the health conditions of healthcare workers during the COVID-19 pandemic. We will certainly use this publication in one of the papers we are preparing on healthcare workers and the pandemic.

Round 2

Reviewer 3 Report

thanks for the corrections made, they are satisfactory